# Soil Fungal Community Composition Correlates with Site-Specific Abiotic Factors, Tree Community Structure, and Forest Age in Regenerating Tropical Rainforests

**DOI:** 10.3390/biology10111120

**Published:** 2021-10-31

**Authors:** Irene Adamo, Edgar Ortiz-Malavasi, Robin Chazdon, Priscila Chaverri, Hans ter Steege, József Geml

**Affiliations:** 1Tropical Botany Research Group, Naturalis Biodiversity Center, 2300 Leiden, RA, The Netherlands; i.adamo90@gmail.com (I.A.); hans.tersteege@naturalis.nl (H.t.S.); 2Faculty of Science, Leiden University, 2300 Leiden, RA, The Netherlands; 3Joint Research Unit CTFC–AGROTECNIO, E-25198 Lleida, Spain; 4Centro de Investigación en Innovación Forestal, Instituto Tecnológico de Costa Rica, Cartago 156-7050, Costa Rica; eortiz2020@gmail.com; 5Department of Ecology and Evolutionary Biology, University of Connecticut, Storrs, CT 06269, USA; rchazdon@gmail.com; 6Escuela de Biología and Centro de Investigaciones en Productos Naturales, Universidad de Costa Rica, San José 11501-2060, Costa Rica; pchaverr@umd.edu; 7Department of Plant Sciences and Landscape Architecture, University of Maryland, College Park, MD 20742, USA; 8ELKH-EKKE Lendület Environmental Microbiome Research Group, Eszterházy Károly Catholic University, H-3300 Eger, Hungary

**Keywords:** biodiversity, Costa Rica, forest communities, impact of disturbance, ITS, Pacific wet forest, succession dynamics, tropical ecology

## Abstract

**Simple Summary:**

Regenerating forests represent over half of all tropical forests. While regeneration processes of trees and animal groups have been studied, there is surprisingly little information about how the diversity and community composition of fungi and other microorganisms change and what ecological roles play in tropical forest regeneration. In this study, we compared the diversity and community composition of trees and soil fungi among primary forests and regenerating forests of different ages in two sampling areas in southern Costa Rica. Our study shows that while forest age has a significant influence, environmental factors, such as mesoclimate and soil chemistry, have stronger effects on both fungal and tree communities. Moreover, we observed that the more dissimilar tree communities are between any two sites, the more dissimilar the composition of fungal communities. The results presented here contribute to a better understanding of the successional processes of tropical forests in different regions and inform land use and forest management strategies, including, but not limited to, conservation, restoration, and sustainable use.

**Abstract:**

Successional dynamics of plants and animals during tropical forest regeneration have been thoroughly studied, while fungal compositional dynamics during tropical forest succession remain unknown, despite the crucial roles of fungi in ecological processes. We combined tree data and soil fungal DNA metabarcoding data to compare richness and community composition along secondary forest succession in Costa Rica and assessed the potential roles of abiotic factors influencing them. We found a strong coupling of tree and soil fungal community structure in wet tropical primary and regenerating secondary forests. Forest age, edaphic variables, and regional differences in climatic conditions all had significant effects on tree and fungal richness and community composition in all functional groups. Furthermore, we observed larger site-to-site compositional differences and greater influence of edaphic and climatic factors in secondary than in primary forests. The results suggest greater environmental heterogeneity and greater stochasticity in community assembly in the early stages of secondary forest succession and a certain convergence on a set of taxa with a competitive advantage in the more persisting environmental conditions in old-growth forests. Our work provides unprecedented insights into the successional dynamics of fungal communities during secondary tropical forest succession.

## 1. Introduction

Secondary forests, i.e., forests regenerating following a major disturbance that altered stand structure, represent over half of all tropical forests, and studying them can offer fundamental insights into the dynamics of community composition, functionality, and ecosystem services along secondary succession [1,2]. Studies on secondary succession in the Neotropics have been focusing mostly on vascular plants and faunal communities [1,2,3,4,5,6,7,8,9,10,11,12,13]. According to the intermediate disturbance hypothesis (IDH), the highest tree species diversity occurs in middle-aged stands transitioning from early to late-successional species [14]. Early-successional stages are characterized by heliophilic plants with rapid growth and high productivity, while slow-growing, shade-tolerant trees that have a competitive edge under limiting resources dominate late-successional stages [15,16]. Mid-successional stands are expected to have representatives of both early- and late-successional species and, therefore, higher tree richness. For example, in Barro Colorado Nature Monument in Panama, tree species diversity was observed to be higher in 20 to 40 years old stands than in older stages of succession [17]. When ant diversity was assessed in secondary forests of the Rio Cachoeira Nature Reserve in Brazil, the recovery progress was shown to be slow, and ant assemblages resembling those in the primary forests only developed after 50 to several hundred years [18]. In a recent study, successional changes in assemblages of amphibian and reptiles were analyzed in the first decades of secondary succession and in primary forests, i.e., forests not subjected to major disturbance, of Southern Mexico. Results reported that amphibian and reptile assemblages rapidly increased in species diversity along the chronosequence, reaching primary forest diversity values in less than three decades [10]. Such studies have provided important insights into successional dynamics, but it is important to consider that community reassembly can deviate from the expected trajectory due to site-specific factors, e.g., soil conditions, initial taxonomic and functional composition, land-use history, and stochastic factors [3,11,19].

Despite the enormous diversity of fungi and their crucial roles in ecological processes as decomposers, mutualists, or pathogens of plants and animals [20,21], little is known about how fungal communities respond to and recover from severe disturbance in forest communities. While successional dynamics of fungi, particularly ectomycorrhizal and wood-decay basidiomycetes, have been studied in northern temperate regions [22,23,24,25], data regarding the changes in diversity and composition of fungal communities during secondary forest succession in the tropics are virtually absent, and only a few studies have investigated the effects of land use on soil fungal communities in the tropic biome [26,27,28]. In this study, we compared the richness and community composition of trees and taxonomic and functional groups of fungi along secondary forest succession in Costa Rica.

We hypothesized that fungal richness would differ through the chronosequence (Hypothesis 1). More specifically, based on previous successional studies on other tropical forest organisms, we formed three alternative hypotheses: fungal richness would be highest in H1a) early-successional stages, as reported in herbs, due to newly available niches created by severe disturbance [3]; H1b) in a mid-successional stand, where representatives of early- and late-successional species are expected to be present [14]; or H1c) in the late stages of succession, as reported in animals and trees, possibly because of the gradual accumulation of microhabitats and shade-tolerant species [1,5,18]. Furthermore, we hypothesized that compositional turnover (β-diversity) would be greater in early-successional stands than either in primary forests or in old-growth secondary forests because of the greater influence of site-specific environmental conditions and the greater role of stochasticity when new habitats are colonized shortly after disturbance (i.e., deforestation) (Hypothesis 2). Conversely, old-growth secondary and primary forests are expected to have more similar communities due to increased environmental filtering and increased competition for more limited resources [14]. Regarding fungal community composition, we hypothesized, based on the above temperate studies, that there would be substantial changes in species composition along the chronosequence as the communities’ transition between early- to late-successional specialists because fungi with different life strategies will respond differently to disturbance and increased competition in young and old-growth forests, respectively (Hypothesis 3). Moreover, we hypothesized that plant-associated fungi and trees would show similar compositional patterns as expected by the Janzen-Connell dynamics, where tree diversity is maintained by specialist pathogens, which reduce the survival rate of conspecific seeds located close to reproductive adults [29,30]. Finally, because soil fungal communities are strongly shaped by environmental factors [31], we hypothesized that regional climate, vegetation composition, and soil characteristics would strongly influence fungal community composition throughout secondary succession (Hypothesis 4).

## 2. Materials and Methods

### 2.1. Study Area

The sampling sites were in the northern and southern parts of the Osa Peninsula, in the Golfo Dulce Forest Reserve, in the province of Puntarenas, Costa Rica. The reserve was established to promote sustainable forest management and biodiversity conservation, and the area hosts secondary forests in different successional stages [32,33,34]. The area of study belongs to the Pacific wet tropical forest biome and constitutes the largest tract of lowland forest on the American Pacific coastline with elevation ranging from sea level to 745 m [35]. The climate of the peninsula is warm and humid, with a mean annual temperature of ~26 °C and mean annual rainfall ranging from 3000 to 7000 mm per year, with a short and not very pronounced dry season between January and March [35]. Due to the orographic formation, the region is characterized by a high plant diversity [36]. The number of vascular plant species within the Golfo Dulce region is almost 2700 [37]. Among trees, plant families with the highest species richness are Fabaceae and Rubiaceae, while Poaceae and Orchidaceae are the richest herbaceous and epiphytic plant families, respectively [37,38]. Moreover, of the 458 tree species collected on the Osa Peninsula, 4.8% were found to be endemic [39]. The peninsula is characterized by different geological formations: the north is dominated by older basalts, while Pliocene sediments make up the central portion, and Quaternary alluvium is present in flood plains [35,40]. Finally, soil types that characterize the peninsula are mainly ultisols, although inceptisols and mollisols are also found in the southern part [35].

### 2.2. Sampling and Molecular Work

The fieldwork was conducted in 2014 October. We sampled nineteen permanent plots (50 × 100 m; between 8°21′07″ N to 8°46′56″ N and 83°14′28″ W to 83°45′24″ W), established by the Instituto Tecnológico de Costa Rica and the University of Connecticut between 2009 and 2011 [34,41,42,43,44]. The plots represent all successional stages and were almost equally divided between the Mogos (northern Osa) and Piro (southern Osa) regions (Appendix A). Four forest types were compared: (i) early-successional secondary young forests between 5 and 15 years of age, (ii) mid-successional secondary forests (15–30 y), (iii) late-successional secondary forests (>30 y) old, and (iv) primary forests. Tree community data of the permanent plots, based on a minimum diameter at breast height (dbh) of 5 cm, were obtained from a vegetation study focusing on the sampled plots [34]. Forty soil cores, 2 cm in diameter and ca. 20 cm deep, were taken in each permanent plot, which were later mixed, and ca. 20 g of composite soil sample was preserved for each plot. Because of the remote location and the lack of facilities, soil samples were first air-dried on the day of collecting and were subsequently placed in sealed bags with silica gel for complete drying. Genomic DNA was extracted at the Universidad de Costa Rica from 0.5 mL of dry soil from each sample using NucleoSpin^®^ soil kit (Macherey-nagel Gmbh & Co., Düren, Germany), according to the manufacturer’s protocol. The internal transcribed spacer 2 (ITS2) region (ca. 250 bp) of the nuclear ribosomal DNA repeat was PCR amplified using primers fITS7 [45] and ITS4 [46]. The ITS4 primer was labeled with sample-specific multiplex identification DNA tags (MIDs). The amplicon library was sequenced at Naturalis Biodiversity Center (Naturalis) using Ion 318TM Chip and an Ion Torrent Personal Genome Machine (Life Technologies, Guilford, CT, USA.). Detailed protocols of molecular work are described in the work of [47]. Soil physico-chemical analyses were carried out at Brookside Laboratories, Inc. (New Knoxville, OH, USA.) to determine pH (H2O), total exchange capacity, organic matter, estimated N release, contents of soluble S, P, available P (Bray II), Ca, Mg, K, Na, H, and other bases, as well as micronutrients, such as B, Fe, Mn, Cu, Zn, and Al.

### 2.3. Bioinformatic Work

The initial clean-up of the raw data was carried out using the online platform Galaxy (https://main.g2.bx.psu.edu/root; accessed on 5 November 2015), in which the sequences were sorted according to samples and adapters (identification tags) were removed. Primers were removed, and poor-quality ends were trimmed with a quality threshold of 20 using Cutadapt 1.14 [48]. Subsequently, quality filtering of the sequences was performed in VSEARCH 2.5.1 [49] based on the following settings: all sequences were truncated to 200 bp, and sequences with expected error >1 were discarded. For each sample, sequences were collapsed into unique sequence types while preserving their counts and setting a minimum count of 2 in order to remove the singletons. Putative chimeric sequences were removed de novo with using VSEARCH 2.5.1 [49]. The quality-filtered sequences from all samples were grouped into operational taxonomic units (OTUs) at 97% sequence similarity using VSEARCH. Sequences were then assigned to taxonomic groups based on pairwise similarity searches against the curated UNITE+INSD fungal ITS sequence database (version released on October 10, 2017), containing identified fungal sequences with assignments to species hypothesis (SH) groups delimited based on dynamic similarity thresholds [50]. After excluding <150 bp pairwise alignment length to a fungal sequence, 2730 fungal OTUs were retained, representing a total of 151,094 quality-filtered sequences.

Only OTUs with >90% similarity to a fungal SH with known ecological function were assigned to one of the following functional groups: parasites (animal pathogens and mycoparasites), plant pathogens, root-associated fungi (mycorrhizal fungi and root endophytes), saprotrophs (generalists), and wood decomposers. The initial functional assignments were made by FunGuild [51] and were manually checked afterward based on ecological metadata of the corresponding SHs in UNITE for genera that are known to comprise species with diverse functions.

### 2.4. Statistical Analyses

Statistical analyses were implemented in the R software environment (version 3.6.0, R Development Core Team 2019, R Foundation for Statistical Computing, Vienna, Austria, URL: www.R-project.org, accessed on 30 September 2021). The iNEXT package [52] was used for fungal diversity analyses, while the vegan package [53] was used for the multivariate analyses.

The OTU table was normalized for subsequent statistical analyses by rarefying the number of high-quality fungal sequences to the smallest library size (5097 reads) using the *rrarefy* function implemented in the *vegan* R package [53]. The resulting matrix contained 2558 fungal OTUs. Linear regression analyses in R were used to test the correlation between fungal richness and abiotic variables. Due to the low number of OTUs assigned to ectomycorrhizal and arbuscular mycorrhizal, these two groups were merged into a combined group of root-associated fungi that also included putative root endophytes. For each taxonomic and functional group and tree species, we used Hill’s diversity N0 [54] to describe the differences in fungal richness values across stand age classes. Hill’s diversity consists of three numbers: N0 is species richness; N1 is the antilogarithm of Shannon’s diversity index; and N2 is the inverse of Simpson’s diversity index. The extrapolated confidence intervals were used to visualize the differences between the stand age classes. The β-diversity was measured as pairwise abundance-based Jaccard similarity [55] among plots within the same stand age class, using probabilistic estimators that have been shown to reduce the negative effects of incomplete sampling of rich communities [54]. The *lmPerm* package [56] was used to run a permutation one-way ANOVA using the function *aovp* to test the effect of the successional stage on β-diversity. Moreover, due to an unequal division of plots for the two regions, additional analyses were performed merging secondary forest stands to detect any difference between primary and secondary forest for Piro vs. Mogos. The function *aovp* was used to compare β-diversity within the successional forest type of the chronosequence. Changes of fungal and tree community composition and fungal and tree families’ proportional relative abundance across environmental variables were tested with canonical correspondence analysis (CCA) in PC-ORD [57]. In addition, permutational multivariate analysis of variance was carried out using the *adonis* function in the *vegan* package [53] to estimate the relative importance of successional stage and region (categorical variables) and edaphic and climatic conditions (continuous variables) as sources of variation in fungal and tree community composition. Climatic data for the sampling locations were downloaded from the WorldClim database (www.worldclim.org, accessed on 4 December 2017). To account for multicollinearity among edaphic variables, correlations were assessed by principal component analysis (PCA), and ordination scores on the first PCA axis were used as combined edaphic index values for downstream analyses. Environmental variables were standardized for mean and standard deviation using the *scale* function in R. Moreover, Mantel tests were performed using the *Mantel* function of the *vegan* package to reveal any spatial autocorrelation in environmental variables as well as in fungal community composition among the sampling sites and to measure the correlation between fungal community and tree community composition as well as between fungal community composition and abiotic variables. In addition, partial Mantel tests were performed to differentiate between the effects of the tree community, the spatial variables, and the abiotic variables on the fungal community structure. The *vegan* package was used to run global nonmetric multidimensional scaling (GNMDS) using the *metaMDS* function on the Hellinger-transformed abundance table and a secondary matrix containing the successional age classes, bioclimatic variables, soil variables, and abundance of tree families. The *envfit* function, implemented in the vegan package, was used to fit the above variables into the fungal ordination. Finally, indicator species in each successional stage were inferred using multi-level pattern analysis with the *multipatt* function in the *indicspecies* R package [58,59].

Total β-diversity was partitioned into replacement (i.e., turnover: the substitution of a species by a different one) and nestedness (where a poor community is the strict subset of a richer one) components. We used Sørensen dissimilarity as total β-diversity and estimated the replacement (Simpson dissimilarity) and nestedness components on presence/absence data using the betapart R package [60].

## 3. Results

### 3.1. Environmental Differences between Regions and Between Primary and Secondary Forests

Mean diurnal temperature range (t = −4.151, *p* = 0.001), mean annual precipitation (t = −11.14, *p* = < 0.001), and precipitation seasonality (t = 4.020, *p* = < 0.001) differed significantly among the two regions, despite the short distance (ca. 30 km) between them. Precipitation and diurnal temperature range were higher in Mogos, while seasonal variation in precipitation was greater in Piro. With respect to edaphic variables, organic matter content (F = 18.177, *p* = <0.001) and estimated nitrogen release (F = 7.748, *p* = 0.013) were significantly greater in Mogos, while Na content was slightly greater in Piro (F = 5.870, *p* = 0.028). Of the measured variables, only Mn content was statistically different between primary and secondary forests (F = 19.325, *p* = 0.0005), while pH did not differ statistically between regions, nor between primary and secondary forests (Appendix A).

### 3.2. Patterns of Fungal Richness

Differences in fungal richness (N0) were detected between primary forests and early (5–15 y), and late-successional secondary forests (>30 y) as the extrapolated confidence intervals did not overlap. We observed the highest richness values in mid-successional stands (15–30 y), but with no significant difference from the fungal richness in primary forests. Tree richness differed significantly along the succession, showing a steady increase with age (Figure 1). In both Mogos and Piro regions, tree species richness was significantly higher in primary than in combined secondary forests. In addition, tree richness was significantly higher in Mogos than in Piro. On the other hand, the fungal OTU richness was significantly higher in Piro than in Mogos, and a significant difference was observed between primary and secondary forests as the extrapolated confidence intervals did not overlap (Figure 1).

When functional groups were analyzed, no differences in richness were detected for plant pathogens, wood decomposers, and root-associated fungi (Figure 2). Conversely, the richness of saprotrophic fungi was lowest in early-successional stands, which was significantly different from that in mid-successional and primary stands. Finally, significant differences were detected between parasitic fungal richness in early-successional and primary forests. With respect to functional groups, OTU richness in saprotrophic fungi was significantly different between the two regions, while significant differences in richness between the two forest types were observed for parasitic and root-associated fungi when the two regions were combined. Significant differences between forest types were only observed in plant pathogens, where richness was highest in secondary forests in the Piro region. (Figure 2). Finally, only parasitic OTU richness correlated positively with tree richness (r^2^ = 0.187, *p* = 0.036) (Figure 3). Significant correlations were found between fungal richness and edaphic variables as well as between tree family abundance and edaphic variables. Total fungal richness (r = −0.585, *p* = 0.008) and the richness of saprotrophic fungi (r = −0.59, *p* = 0.008) correlated negatively with organic matter. Estimated nitrogen release was negatively correlated with total fungal richness (r = −0.51, *p* = 0.025), and with the richness of wood-decay fungi (r = −0.408, *p* = 0.034). On the other hand, total exchange capacity was positively correlated with the richness of plant pathogenic fungi (r = 0.478, *p* = 0.038) and parasitic fungi (r = −0.458, *p* = 0.0482). In addition, the richness of parasitic fungi was also positively correlated with pH, K and Ca (r = 0.469, *p* = 0.042; r = 0.642, *p* = 0.002; r =0.504, *p* = 0.027, respectively).

### 3.3. Community Composition and β-Diversity

Fungal β-diversity was significantly lower in primary than in early- and mid-successional forests (*p* = 0.002), with intermediate measures in late-successional secondary forests (Figure 1). Tree species β-diversity was also lowest in primary forests and differed significantly from that in the secondary forests of various successional stages (*p* = 0.0259). The above total fungal β-diversity pattern was also seen in plant pathogenic (*p* < 0.001) and, to a lesser extent, in saprotrophic (*p* = 0.009) and wood-decay fungi (*p* = 0.022), while no significant difference in β-diversity was observed in parasitic and root-associated fungi along the chronosequence (Figure 2). Furthermore, both tree and fungal β-diversity measures were statistically different between primary and secondary forests (*p* < 0.001; *p* = 0.005). Moreover, the fungal and tree β-diversity was lower in the primary than the secondary forests in both regions (Figure 1). At the functional level, β-diversity were significantly different between forest types in plant pathogenic (*p* = 0.007) and in wood-decay fungi (*p* < 0.001), while β-diversity of saprotrophic fungi significantly differed between forest types and geographic regions (forest type: *p* = 0.018; region: *p* = 0.045) (Figure 2). In addition, the parasite β-diversity was not significantly different between forest types or regions. Similar to observed α-diversity patterns, Tukey’s HSD tests indicated that the differences in β-diversity between primary and combined secondary forests were only significant in Mogos. There, all functional groups, except parasites, showed significantly lower β-diversity in primary than in secondary forests (Figure 2). Finally, the only significant interaction between forest type and region was observed for root-associated fungi (*p* = 0.007) (Figure 2). In addition, saprotroph, plant pathogens and wood-decay fungi β-diversity correlated positively with tree species β-diversity (Mantel r = 0.327, *p* = 0.001, r = 0.492, *p* = 0.001, r = 0.4, *p* = 0.001) (Figure 3). Multi-level pattern analysis revealed the highest number of indicator fungal OTUs for the primary forest (15 OTUs), while the number of indicator OTUs for the secondary forest age groups varied between 1 and 4 (Appendix A).

For the fungal data set, the first axis of the CCA explained the greatest variation (eigenvalue = 0.378, *p* = 0.01). Forest age, precipitation seasonality, mean diurnal temperature range negatively correlated with the first axis, while mean annual precipitation was positively correlated. Proportional relative abundance of Botryosphaeriaceae (plant pathogens), Trichocomaceae, Trichosphaeriaceae, and Aspergillaceae (saprotrophs) was higher in older stands (negative first-axis values in Figure 4), while the average relative abundance of Clavicipitaceae (parasites), Nectriaceae and Sporocadaceae (plant pathogens) was higher in wetter sites (positive first-axis values in Figure 4), and positively correlated with organic matter, Al and estimated nitrogen release (Figure 4 and Figure 5). On the other hand, the average relative abundance of Xylariaceae (wood-decay fungi) and Ganodermataceae (plant pathogens) correlated with higher pH, total exchange capacity, P, Na, and Fe, while it was negatively correlated with mean annual precipitation. Finally, the proportional relative abundance of Chaetomiaceae and Herpotrichiellaceae (saprotrophs) was higher in early-successional stands (positive first-axis values in Figure 4). 

With respect to the tree data set, the first axis of the CCA explained the greatest variation (eigenvalue = 0.423, *p* = 0.001). Here, the proportional relative abundance of Burseraceae, Clusaceae of Moraceae, Myristicaceae, Lauraceae, and Anarcadiaceae was higher in late-successional stages and drier sites (negative first-axis values in Figure 4), while the average relative abundance of Flacourtaceae, Melastomaceae, Euphorbiaceae, and Vochyisiaceae was higher in wetter sites (positive first-axis values). Finally, the proportional relative abundance of Tiliaceae, Fabaceae, and Rubiaceae correlated with soil variables (positive first-axis values), therefore with high values pH, total exchange capacity, P, Na, and Fe (Figure 4 and Figure 5).

### 3.4. Effects of Environmental Factors on Fungal Community Composition

Considering all plots, stand age explained the greatest proportion of variation in both fungal (26.96%, *p* = 0.007) and tree communities (31.72%, *p* = 0.002), followed by edaphic variables (15.33%, *p* = 0.003, 22.63%, *p* = 0.001), mean annual precipitation (14.40%, *p* = 0.005, 26.22%, *p* = 0.001) and precipitation seasonality (13.42%, *p* = 0.01, 25.47%, *p* = 0.001), with the explained variation being generally greater in trees (Table 1). The effects of forest age, soil variables and climate were complementary, because their contributions to explained variation remained significant in the combined model. In addition, edaphic variables (21.90%, *p* = 0.002, 32.7%, *p* = 0.001), mean annual precipitation (19.82%, *p* = 0.007, 31.36%, *p* = 0.001) and precipitation seasonality (18.52%, *p* = 0.008, 30.59%, *p* = 0.001) strongly influenced fungal and tree community composition in secondary forests. On the other hand, fungal communities in primary forests were primarily driven by precipitation seasonality (32.38%, *p* = 0.05) and diurnal temperature range (30.39%, *p* = 0.03), while tree communities were primarily driven by mean annual precipitation (78.34%, *p* = 0.017) and precipitation seasonality (66.4%, *p* = 0.017).

With respect to functional groups, successional stage (r^2^ = 0.390, *p* = 0.014, r^2^ = 0.376, *p* = 0.027) strongly influenced community structure in saprotrophic and wood-decay fungi, while soil characteristics (r^2^ = 0.514, *p* = 0.002, r^2^ = 0.608, *p* = 0.001) had a significant effect on plant pathogens and saprotrophs. Mean annual precipitation and precipitation seasonality (r^2^ = 0.346, *p* = 0.033, r^2^ = 0.400, *p* = 0.020) strongly influenced saprotrophic fungal composition, while mean diurnal temperature range (r^2^ = 0.511, *p* = 0.006, r^2^ = 0.511, *p* = 0.006) had a significant effect on saprotrophic and wood-decay fungal community structure. Finally, mean temperature diurnal range (r^2^ = 0.315, *p* = 0.039) had a significant effect on parasitic fungal composition, while neither soil nor bioclimatic variables influenced root-associated fungal community structure (Appendix A). Abundance of several trees influenced the composition of the functional groups. Of these, (r^2^ = 0.427, *p* = 0.006), Flacourtaceae (r^2^= 0.408, *p* = 0.025), Sterculiaceae, and Euphorbiaceae (r^2^ = 0.337, *p* = 0.039) had a significant effect on saprotroph community composition, Rubiaceae, Vochysiaceae (r^2^ = 0.322, *p* = 0.045, r^2^ = 0.429, *p* = 0.024), had a significant effect on saprotrophs and parasites community composition, while Fabaceae (r^2^ = 0.475, *p* = 0.008, r^2^ = 0.645, *p* = 0.001), Rubiaceae (r^2^ = 0.387, *p* = 0.022, r^2^ = 0.309, *p* = 0.036), Meliaceae (r^2^ = 0.403, *p* = 0.022, r^2^ = 0.461, *p* = 0.008) and Clusiaceae (r^2^ = 0.401, *p* = 0.018, r^2^ = 0.369, *p* = 0.014) had a significant effect on saprotroph and wood-decay fungal community composition. Moreover, community composition of wood-decay fungi was strongly correlated with Tiliaceae (r^2^ = 0.461, *p* = 0.005), Annonaceae (r^2^ = 0.675, *p* = 0.001) and Burseraceae (r^2^ = 0.420, *p* = 0.004), and plant pathogenic fungal community structure with Melastomaceae (r^2^ = 0.410, *p* = 0.029).

Mantel tests revealed tree community composition correlated spatially (r = 0.284; *p* = 0.002) and with edaphic variables (r = 0.339; *p* = 0.004), even though no spatial correlation was detected in the latter (r = −0.072; *p* = 0.72). Fungal community composition correlated most strongly with edaphic variables (r = 0.439; *p* = 0.002) and tree community composition (r = 0.293; *p* = 0.002) and showed much weaker, though still significant, spatial correlation (r = 0.198; *p* = 0.037). Partial Mantel tests revealed that regional effect on fungal and tree communities were primarily driven by climatic differences, because neither fungal (r = 0.121; p = 0.168), nor tree communities (r = 0.109; *p* = 0.171) showed significant spatial structure when climatic variables were controlled. Partial Mantel tests revealed that when the effect of tree community composition was controlled for, spatial correlation of the fungal community was not significant (r = 0.136; *p* = 0.106), while the effect of edaphic variables remained significant (r = 0.333; *p* = 0.01). Conversely, the correlation between tree and fungal community always remained strong (r = 0.25; *p* = 0.016, r = 0.225; *p* = 0.013) when the effect of the spatial or edaphic variables was controlled for.

When β-diversity of fungi was partitioned into replacement and nestedness components, we observed that replacement accounted for most of observed beta diversity, with nestedness being small (Figure 6). Linear regression analyses indicated significant positive relationships between pairwise differences in replacement and all groups environmental variables, with the greatest correlation observed between fungal β-diversity and differences in edaphic factors, while nestedness did not correlate with any. Regarding tree communities, the dominance of replacement was also evident. Nevertheless, significant correlations with pairwise differences in environmental variables were observed both for replacement and nestedness among samples. Tree community turnover correlated positively with spatial distance as well as with edaphic and climatic differences among the sampling sites. Tree community nestedness was negatively correlated with edaphic and climatic differences and showed a positive correlation with age differences (Figure 6).

## 4. Discussion

The results from our pioneering study, albeit based on a limited number of well-characterized permanent forest plots, clearly show the coupling of tree and soil fungal community structure in wet tropical primary and regenerating secondary forests, with some important differences. More specifically, both tree and fungal communities appear to be shaped by forest type (primary vs. secondary) and differences in climatic and edaphic variables. Although the relationship between vegetation and fungal community composition confirms the importance of plant-fungal interactions in shaping tropical forest structure [61,62,63], ours is perhaps the first study to document the strong influence of regional differences in edaphic and climatic factors on the successional dynamics of tropical rainforest trees and fungi, including distinct successional patterns among various functional and taxonomic groups. In addition, we also observed greater environmental heterogeneity and greater stochasticity in community assembly of both trees and fungi in the early stages of secondary forest succession and a certain compositional convergence in the more persisting and relatively more stable environmental conditions in old-growth forests, as we discuss below in detail.

We found evidence for marked changes in fungal richness between primary and regenerating secondary forests, similar to various patterns of richness reported for an array of other organismal groups in primary and regenerating secondary tropical forests [6,8,19,64,65]. However, fungal richness appeared to be highest in mid-successional (15-30 y) forests, likely due to the presence of both early- and late-successional species, which disproves our hypothesis H1a and supports the alternative H1b. Moreover, these significant differences in soil fungal richness differed from what was found in a comparative study of soil fungal communities in pastures and secondary and primary forests in the Brazilian Amazon [24] and during post-fire secondary succession in boreal forests [28,66]. Tree and fungal richness generally did not correlate, which agrees with fungal and plant richness patterns observed globally [21] and in other Neotropical forests [26,67].

In contrast to total fungal richness, plant pathogens and parasites were significantly less diverse in the combined secondary forest plots than in the primary forests. This pattern mirrors that of tree species richness, which was significantly higher in primary than in secondary forests. This difference between primary and secondary forests is similar to what had been observed in other tropical secondary forests [1,2,4,8,19]. However, these studies reported that tree species richness can rapidly recover to values similar to primary forests in about three decades, while in our plots, even late-successional secondary forests had significantly fewer tree species than primary forests. It is noteworthy that both tree and fungal richness were strongly influenced by geographic region (Mogos vs. Piro), despite the relatively short distance and high habitat connectivity between the two regions. Even though both Mogos and Piro are classified as wet Pacific tropical forest, there are some environmental differences between the two regions. The southern region, Piro, is slightly drier and more seasonal, while in the northern region, Mogos, precipitation is higher with minimal seasonal variation [30,35]. Precipitation is a well-known abiotic variable with a strong positive influence on fungal diversity on a global scale [21]. Interestingly, in the study area, while tree richness in the primary forests was significantly higher in the wetter region, Mogos, fungal richness was higher in the slightly drier region, Piro. Moisture does not seem to be a limiting factor in Piro, as it still receives more than 3500 mm annual precipitation, and it is possible that soils are periodically more waterlogged (potentially anoxic) in Mogos, with more than 4200 mm per year, which could result in less favorable conditions for soil fungi. Although we found no spatial autocorrelation in the measured soil chemical properties (Appendix A), several abiotic factors were statistically different between the two regions (Appendix A). Soil organic matter and estimated nitrogen release were higher in the wetter region, while pH, Na, and Mg appeared higher in the drier region; thus, the two regions showed differences in climate as well as in some edaphic factors. While the observed trends were similar, the significant differences in tree, plant pathogen, and parasite richness between primary and secondary forests were primarily driven by the differences observed in Mogos, confirming the importance of regional effects on successional dynamics as discussed in detail below (Figure 1 and Figure 2).

The results presented here support our Hypothesis 2 of significant changes in fungal β-diversity across forest succession. The greater compositional turnover of fungi among plots in early- and mid-successional stands than in primary forests may be explained partly by higher stochasticity in colonization following disturbance and by greater variation in and influence of edaphic variables on composition. In addition, the observed significantly positive correlation between fungal and tree community turnover appears to confirm the strong linkage between soil fungi and vegetation. Similarly, the lower fungal β-diversity measures in the primary forest indicate more similar communities, likely because of environmental filtering under relatively stable conditions over many years, when compared to the initial stages of secondary forest succession [11]. Mature forests likely provide a certain buffer against spatial and temporal differences in environmental factors that can persist for decades or centuries. Therefore, the resulting increased environmental filtering and competition for more limited resources [14] may drive some compositional convergence in primary forests. This is also supported by the multi-level pattern analysis, where the primary forest had by far the highest number of indicator fungal OTUs, i.e., diagnostic and “faithful” taxa. This finding is of crucial importance for conservation because our study suggests that deforestation predictably leads to profound compositional changes and the loss of species that are specifically adapted to primary forests and that these disturbance effects are still apparent after more than 30 years in both tree and fungal communities.

In our study, functional groups of fungi showed significant differences in richness and composition across the four stand ages, which confirms our Hypothesis 3. Moreover, it is worth noting that even within functional groups, many fungal families differed in relative abundance among the forest successional stages and regions, correlating positively or negatively with forest age and environmental variables (Figure 3). For example, the families Trichocomaceae and Trichosporanaceae (saprotrophs) seemed to prefer old-growth forests, while Chaetomiaceae and Herpotrichiellaceae (saprotrophs) occurred mainly in early-successional stages. Similar differences among successional stages were also found in the sporocarp-based study of [68], who found that the diversity of hypocrealean fungi (order Hypocreales, Ascomycota) decreased significantly across secondary forest succession in Sarapiguí, Costa Rica, probably as a result of less decaying plant material and increased competition, leading to less colonization and lower diversity of fruiting structures in older stands. Moreover, in our study, Nectriaceae and Sporocadaceae (plant pathogen) were more abundant in wetter sites, while the family Ganodermataceae (plant pathogen) was more abundant in drier sites. In addition, the families Clavicipitaceae and Cordycipitaceae (parasites) were more prevalent in wetter sites, possibly related to insect host abundance or other unknown factors. We argue that there are clear differences in habitat preference even within functional groups and knowing more about the ecology and natural history of the organisms is fundamental for better understanding patterns of distribution at local and regional scales.

While both fungal and tree communities differed among successional stages, the strong influence of environmental variables of community structure was evident not only for fungi but for trees as well, which agrees with our Hypothesis 4. Both fungal and tree communities were characterized by high community turnover, as a replacement was the dominant component of β-diversity in both organismal groups. This suggests that there are few generalist taxa that are present in most of our sites. Such a high replacement or turnover could reflect either narrow niches and strong environmental filtering, strong interspecific competitive exclusion, or historical contingencies with founder effects allowing early colonizers to limit the establishment of latecomers. This latter may be particularly important in the early stages of secondary succession when stochastic processes are expected to contribute more to community assembly. Without additional data, it is impossible to determine in our data set what the major cause of such a strong community turnover was.

The strong correlation observed between fungal and tree communities and with differences in abiotic factors among sampling sites suggests that fungi and trees respond similarly to changes in environmental factors at the landscape scale (Figure 4, Figure 5 and Figure 6 and Appendix A, Appendix A). Edaphic variables were not only important for soil fungal community composition, as expected, but also for the composition of tree communities (Figure 6), a relationship that is much less documented. The role of soil attributes as one of the main drivers of tree community assembly along forest regeneration is only recently being recognized (e.g., the work of [69]). Moreover, the strong correlation between fungal and tree community composition observed in our study is in agreement with patterns reported from the Amazon [62,67]. We argue that the correlation observed between tree and fungal communities can be explained by a combination of the following mechanisms: (1) direct plant-fungal interactions (e.g., mutualistic mycorrhizal, antagonistic pathogenic, and substrate-specific wood-decay fungi that generally involves certain levels of specificity from the host as well as from the fungus; (2) indirect effect of plants on fungal community structure via altering soil structure, pH and nutrient levels (e.g., via differences in chemical composition leaves among trees); and (3) correlational, but not causal relationships when the environmental niches of certain trees and fungi overlap (habitat specificity) without clear interactions among them. Disentangling the above relationships is beyond the scope of observational studies such as ours and the vast majority of forest metabarcoding studies because it would require experimental setups (e.g., transplanting experiments, litter chemistry measurements, etc.])n specifically designed to test appropriate hypotheses with a wide range of tree and fungal taxa. While such experiments would be logistically immensely challenging already, the biggest obstacle is the current paucity of information on the taxonomic and functional diversity of fungi and their compositional dynamics in tropical forests. Our knowledge on what drives fungal diversity and community composition in the tropics is still very limited compared to temperate regions [21,34,70,71,72]. Nevertheless, the fact that both richness and β-diversity of plant pathogenic and β-diversity of root-associated and wood decomposer fungi mirrored richness and β-diversity patterns of trees suggests that at least in fungi with direct interactions with plants, community structure tends to be coupled with that of trees. For example, specialist pathogens are known to contribute to high tree diversity by causing density-dependent mortality, as explained by the Janzen–Connell dynamics [34,73,74,75]. Though not directly tested in this study, our findings agree with patterns expected under the Janzen–Connell dynamics. We conclude this because the richness and β-diversity (community turnover) positively correlate between hosts and plant pathogens, suggesting some degree of host specificity, and we also observed patchy distribution (e.g., the high plot to plot variation), which is necessary for the Janzen–Connell dynamics to be effective.

## 5. Conclusions

Understanding diversity patterns and community compositional dynamics in regenerating secondary forests is fundamental, as they represent over half of the remaining tropical forests, with a further expected increase in proportion [76]. Our study is one of the first to provide a detailed picture of changes in fungal community structure along secondary forest succession in the tropics. Our study suggests that ecological processes at regional and local (stand) scales are strong enough to shape community composition despite the high diversity and spatial heterogeneity among the sampling sites. In addition to the effects of forest succession on fungal and tree communities, a key finding of our study is that abiotic conditions are strong drivers of successional trajectories. Therefore, more studies are needed from a wide range on tropical forests to better understand the successional dynamics of tropical forest fungi in different regions and to inform land use and forest management strategies, including, but not limited to, conservation, restoration, and sustainable use.

## Figures and Tables

**Figure 1 biology-10-01120-f001:**
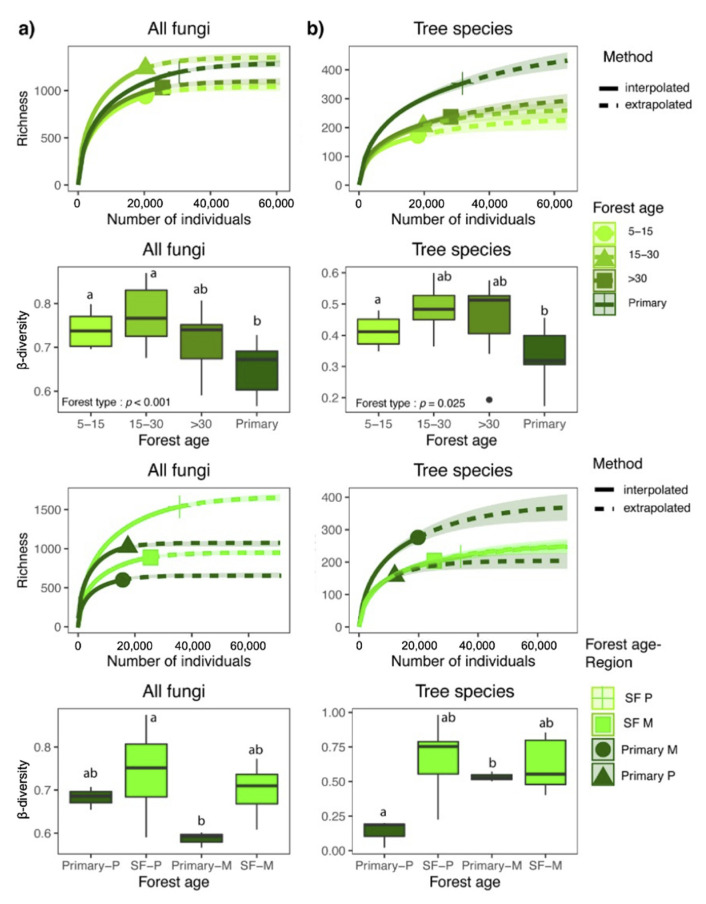
(**a**) Hill’s N0 interpolated and extrapolated richness across different stand age classes. Unbroken and dashed parts of the curve denote interpolated and extrapolated values, respectively, and the shaded zone around each curve denotes the 95% confidence intervals. Comparison of compositional turnover (β-diversity) of fungal and tree communities of early- (5–15 y), mid- (15–30 y), and late-successional (> 30 y) and primary forests based on normalized matrices. (**b**) Hill’s N0 interpolated and extrapolated richness across different stand age classes and compositional turnover (β-diversity) of fungal and tree communities in secondary forests (SF) and primary forests and between geographic regions, Piro (P) and Mogos (M). β-diversity was calculated as an abundance-based Jaccard similarity index. Means were compared using ANOVA and Tukey’s HSD tests, with letters denoting significant differences within each boxplot.

**Figure 2 biology-10-01120-f002:**
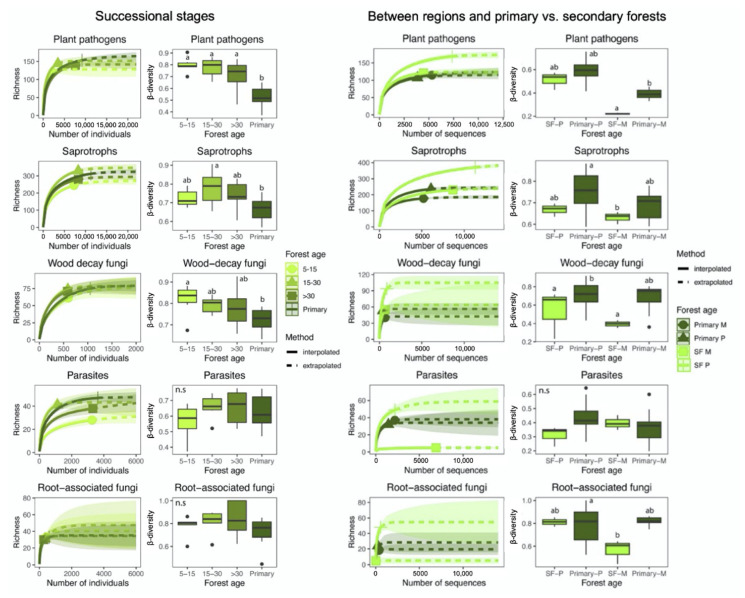
Hill’s N0 interpolated and extrapolated richness values of fungal functional groups across different stand age classes in both regions combined and in primary and all secondary forests (SF) between geographic regions two regions, Piro (P) and Mogos (M). Unbroken and dashed parts of the curve denote interpolated and extrapolated values, respectively, and the shaded zone around each curve denotes the 95% confidence intervals. Boxplots show compositional turnover (β-diversity) of the various functional groups among samples within forest types, calculated as abundance-based Jaccard similarity index. Means were compared using ANOVA and Tukey’s HSD tests, with letters denoting significant differences within each boxplot.

**Figure 3 biology-10-01120-f003:**
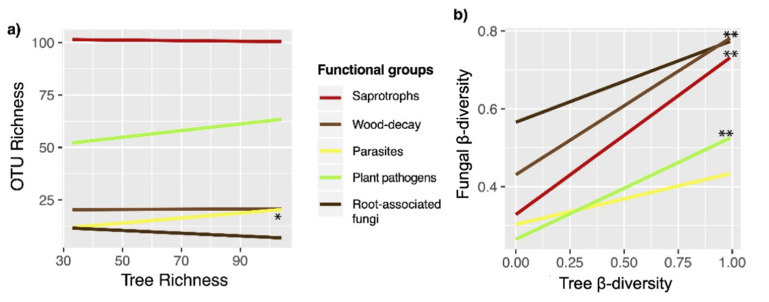
(**a**) Correlation between fungal OTU richness and tree richness (**b**) correlation between fungal β-diversity and tree β-diversity across primary and regenerating secondary forest sites in Costa Rica. Significance levels of correlation are: * = *p* ≤ 0.05 and ** = *p* ≤ 0.01.

**Figure 4 biology-10-01120-f004:**
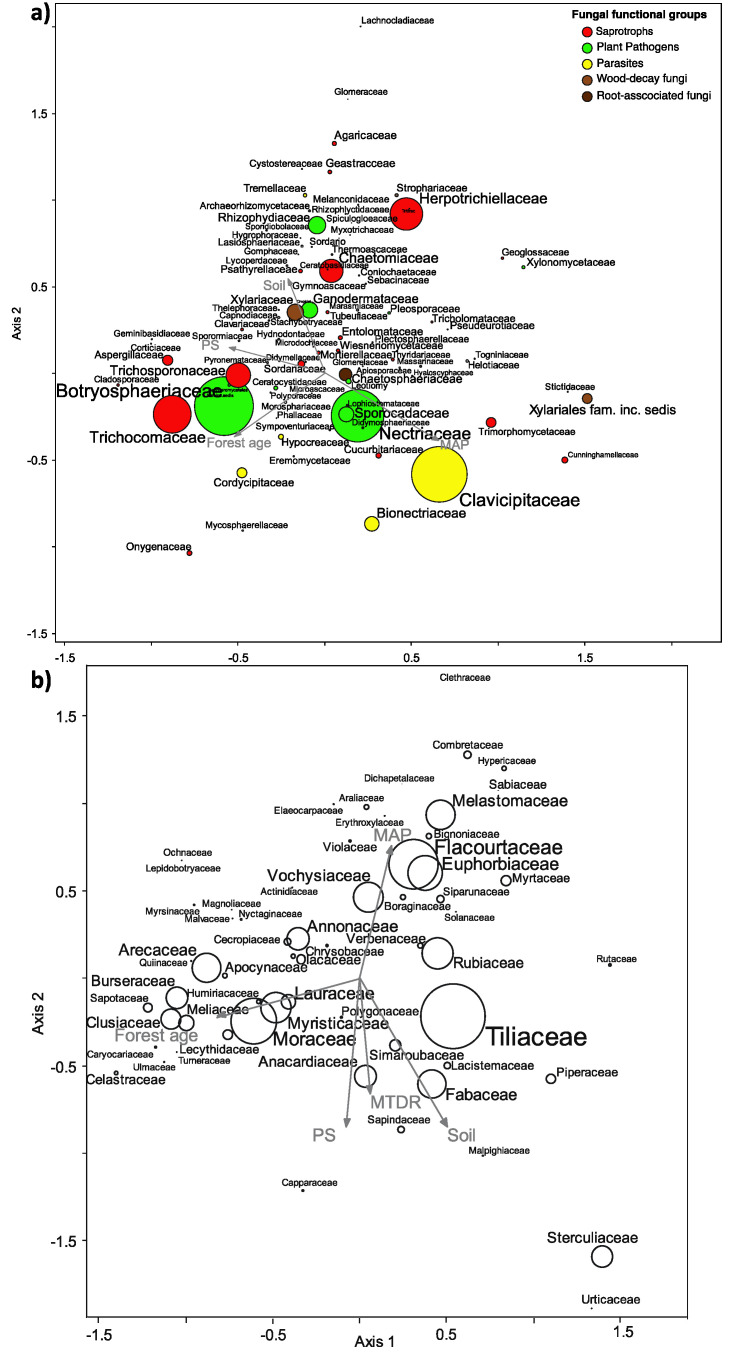
Canonical correspondence analysis (CCA) plots showing the variation in the community composition of (**a**) soil fungi and (**b**) trees using arcsin-transformed relative abundance matrix. Symbols are colored according to functional guilds, and symbol sizes are proportional to the average relative abundance. Abbreviations: MAP: mean annual precipitation, MDRT: mean diurnal range of temperature, PS: precipitation seasonality.

**Figure 5 biology-10-01120-f005:**
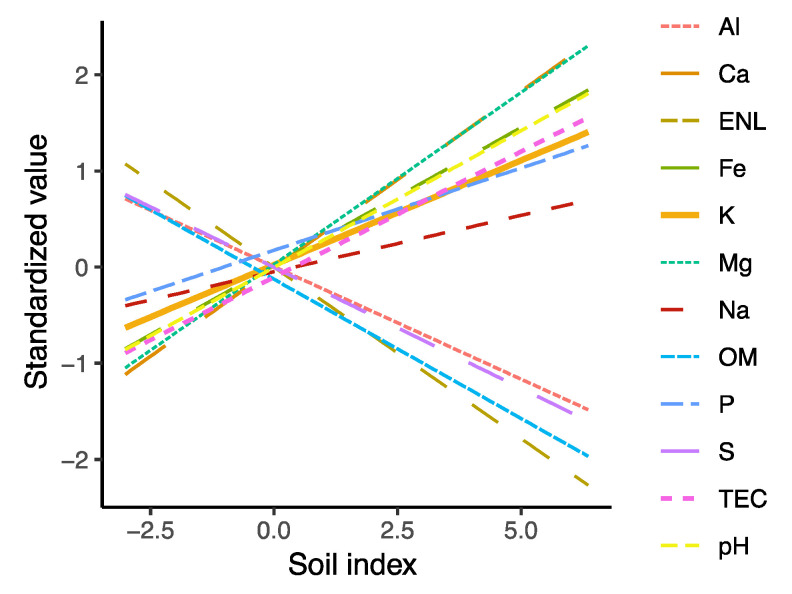
Correlation of soil chemical variables with soil index. Soil index is represented by an index based on ordination scores on the first axis of a principal component analysis (PCA) of all environmental variables.

**Figure 6 biology-10-01120-f006:**
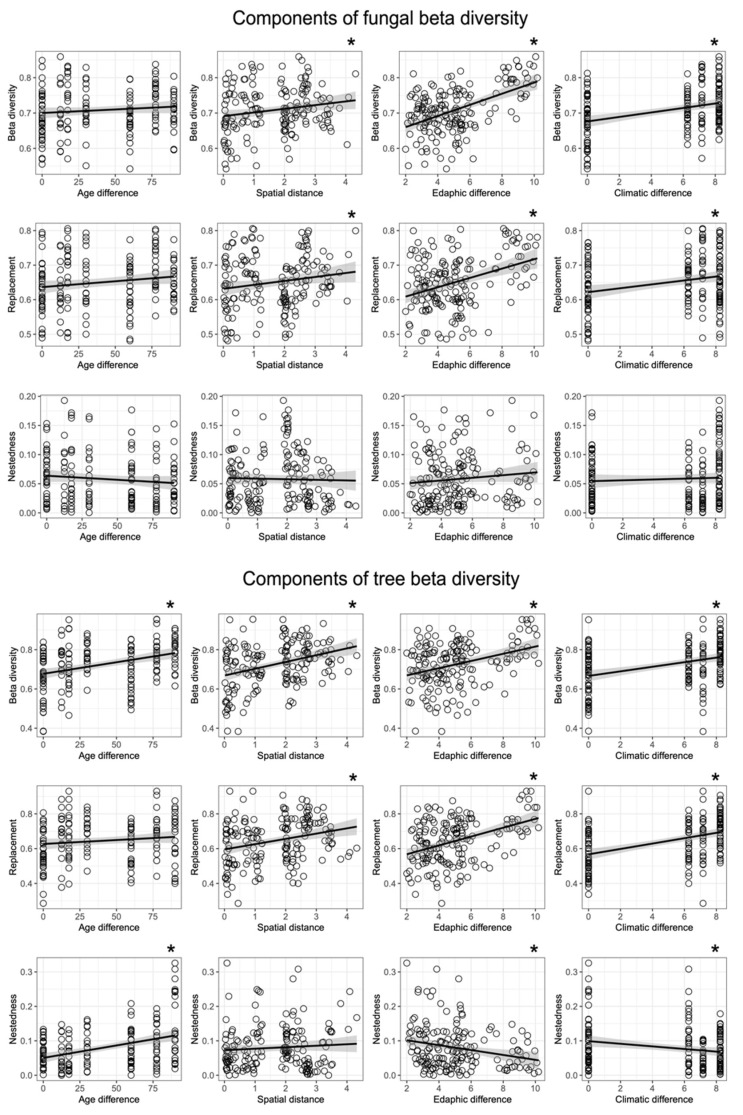
Correlation of total β-diversity and its replacement and nestedness components fungal and tree communities with pairwise differences in environmental factors among the sites. Significant correlations are indicated by asterisk *.

**Table 1 biology-10-01120-t001:** Proportion of variation in fungal and tree community composition explained by successional stage and region (categorical) and edaphic and climatic (continuous) variables as explained in the text. Abbreviations: MAP: mean annual precipitation, MDRT: mean diurnal range of temperature, PS: precipitation seasonality.

Fungal Community	All Sites		Primary Forest		Secondary Forest	
	Var. (%)	*p*	Var. (%)	*p*	Var. (%)	*p*
Successional stage (cat.)	26.96	0.007	-	-	-	-
Region (cat.)	14.40	0.004	25.59	0.3	19.79	0.008
Edaphic index	**15.33** ^a^	**0.003**	17.92	0.659	**21.90**	**0.002**
Age	15.20	0.003	**-**	**-**	-	-
MAP	**14.40**	**0.005**	25.91	0.217	**19.82**	**0.005**
PS	**13.42**	**0.01**	**32.38**	**0.03**	**18.52**	**0.008**
MDRT	**10.5**	**0.003**	**30.39**	**0.05**	15.11	0.056
**Tree Community**	**All Sites**		**Primary Forest**		**Secondary Forest**	
	**Var. (%)**	** *p* **	**Var. (%)**	** *p* **	**Var. (%)**	** *p* **
Successional stage (cat.)	31.72	0.021	-	-	-	-
Region (cat.)	25.98	0.001	78.06	0.300	30.25	0.001
Edaphic index	**22.63**	**0.001**	14.36	0.535	**32.07**	**0.001**
Age	**20.85**	**0.001**	**-**	**-**	**-**	**-**
MAP	**26.22**	**0.001**	**78.34**	**0.017**	**31.36**	**0.001**
PS	**25.47**	**0.001**	**66.4**	**0.017**	**30.59**	**0.001**
MDRT	12.59	0.036	23.58	0.350	18.27	0.042

^a^ Correlations that remained significant in the combined model are in bold.

## Data Availability

DNA sequences have been deposited in the NCBI database under accession numbers MH810434-MH812991.

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
