# Peer review of "Soil Fungal Community Composition Correlates with Site-Specific Abiotic Factors, Tree Community Structure, and Forest Age in Regenerating Tropical Rainforests"

_biology, 2021, doi:10.3390/biology10111120_

Round 1
Reviewer 1 Report
The work by Dr. Adamo and collaborators is a well-written article that employs state-to-the-art methods to combine tree data and soil mycobiome and compare richness and community composition along secondary forest succession in Costa Rica, to further identify potential factors influencing them. The study is interesting and may stimulate various future research in the field of microbiology.
Here are some points that should be checked before publication.
Line 83-85: this sentence should be move to methods
Line 104: “iii) we hypothesized” and remove “iii)” in line 105
Line 175-177: this should be placed in a “Data availability” statement at the end of discussion
Line 445-447: I would restructure this sentence. An example: “The results from our pioneering study, albeit based on a limited number of well-characterized permanent forest plots, clearly show coupling of tree and soil fungal community structure in wet tropical primary and regenerating secondary forests, with some important differences.”
Author Response
Thank you for your constructive comments. We revised the manuscript according to your suggestions. Detailed responses:
Line 83-85: we moved this sentence to methods
Line 104: we removed “iii)" in line 105
Line 175-177: Because the sentence on the accession numbers of the DNA sequence was already in the “Data availability” statement at the end of discussion, we removed the same sentence from here.
Line 445-447: We have restructured the sentence as suggested.
Reviewer 2 Report
Overall impression
This is an interesting study with insights to previously unmapped areas. I like how the importance of this pioneering study is explained. The execution is flawless. The statistical analyses and their presentation is up-to-date; the presence of coloured figures make the manuscript stand out among others. However, there are some minor flaws.
Language
Excellent, with some minor typos and grammatical issues.
Title
The Title is somewhat long, but reflects the content and is relevant.
Authors’ list and affiliations
The affiliation of the last author has a typo in the abbreviated name of the institution.
I think the abbreviated names should be added for an easy guide through the Authors Contribution section.
Simple Summary
To be honest, this is the first time I ever review a “Simple Summary” in an MDPI manuscript during the review process. I have not found any guidelines regarding its content, style or even its length. So I go blindly, and create my opinion based on the name of the section “simple summary”.
This Simple Summary is simple, and conveys the most important information in an easily understandable manner.
I suggest the Authors replace “know” (Line 20) with “knowledge” or “information” or anything to that effect, because although “know” is may be a noun, its use as such is extremely rare, so it goes beyond the limitations of “being simple”.
Also, please finish the last sentence with a full stop (Line 30).
Abstract
The Abstract is within the word count limit. The sections of the article (Introduction, Methods, Results, Conclusions) are well represented.
I think I may have detected a missing preposition in the last sentence in Line 44.
Keywords
There are only 5 keywords, whereas 10 are allowed.
The present keywords are relevant or somewhat relevant, but I would like to encourage the Authors to expand the list, and have 10. I think some of the following suggested keywords may be considered as an addition: “succession”, “succession dynamics”, “forest communities”, “fungal richness”, “fungal species composition”, “soil characteristics”, “beta-diversity”, “impact of disturbance”, “buffer effect”.
The abbreviation “ITS” is not explained in the manuscript, and while it may be a well-known abbreviation for those inside the study field, it remains a mystery for an academic reader coming from another scientific area.
The Authors may consider replacing “Osa peninsula” with “Costa Rica”, as a more general keyword.
Introduction
The terms primary forest and secondary forest are not explained. These expressions are a bit self-explanatory, and anyone with a basic degree in agriculture/biology/ecology may understand them, but I still recommend the Authors add a brief (!) definition of these.
Line 59 The sentence starting as “For example, [17], examined 59 the patterns and processes” is a weird way to cite a literature. I think the proper way would be: „When Denslow et al. [17] examined the patterns and processes…” OR… „The patterns and processes of xxx were examined [17], and they found…”
Apart from that, the Introduction defines the significance of the study, and includes more hypotheses than one may imagine. There are 4 hypotheses, but not all are introduced with the same way. Some have simple brackets, others box brackets. It appears that hypothesis No. 1. has a subhypothesis names “1a”, but there is no “1b” or “1c”.
I suggest the Authors decide upon a method and stay consistent.
Materials and Methods
Overall, the description of the study area, sampling conditions and molecular works are sufficient enough so that other researchers may replicate a similar experiment.
Some minor problems:
Line 120
I don’t think anything justifies to display annual rainfall this fashion “rainfall ranging from 3000 to 7000 mm yr-1 with” I suggest the Authors edit this to “rainfall ranging from 3000 to 7000 mm per year with”. This is not a figure, table, formula or equation. This is a flowing text.
The same mistake appears in Line 508.
The sentence in Lines 123-125 suggests that Fabaceae and Rubiaceae are families with tree members only, which is clearly not the case. Also, families Poaceae and Orchidaceae are not “plants”, as the grammatical composition of the sentence suggests. These can be addressed “families”, or “taxa” (the plural of taxon). I recommend the Authors revise this sentence for clearer adherence to botanical nomenclature.
Line 129 there is one too many comma in the sentence. Erase the one between “peninsula” and “are”
Line 189 is in bold for what reason?
Results
The results are presented in a clear manner.
At the same time however, I am unsure whether experimental (preliminary) conclusions are drawn here from the results.
Figure 4
I may be wrong, but I can’t seem to find explanation to axes 1 and 2 in either of the subfigures.
Discussion
The results and conclusions are discussed in a broader scientific context.
I would have welcomed the reappearance of all the hypotheses described in the Introduction. I would like to ask the Authors to refer to all the hypotheses in their original order and state whether these were confirmed or not. I think the reader deserves a definite “proven”, “not proven” or “doubtful” verdict of the interesting hypotheses. I would like to suggest this alteration even if it involves a rearrangement of passages and/or the creation of subsections within the discussion.
The manuscript and thus, the scientific community would benefit from such a change, although I admit this is not something the Authors are able to do within a couple of minutes. But it sure worth the tiresome revision.
Conclusions
Solid conclusions drawn from the study and its results.
Line 596 may be missing a definite article “the”.
References
The number of references is satisfactory. The number of self-references is low, only 7 are detected.
The Manuscript relies on up-to-date literary references, most of the 77 references date in the XXI. century.
Reference #70: The first word of the title (“Soil-“, yes, with a hyphen) is missing.
Reference #67: I am not sure if this is cited correctly.
When I looked up, this citation is what I came up with:
Taylor, D. Lee; Herriott, Ian C.; Stone, Kelsie E.; McFarland, Jack W.; Booth, Michael G.; Leigh, Mary Beth. 2010. Structure and resilience of fungal communities in Alaskan boreal forest soils. Canadian Journal of Forest Research. 40: 1288-1301.
Source: https://www.fs.usda.gov/treesearch/pubs/39056
Please check and correct, if needed.
Reference #45: This is not in English. Please provide a concise translation in box brackets. The same applies to all non-English titles.
Reference #44: There is something going on here. Some names are not capitalized; some names are doubled. Please correct. Also, this is not in English. Please provide a concise translation in box brackets.
Reference #39: Are you sure the first name of Dr Hietz starts with a “G”? This is what I have found: https://www.zobodat.at/pdf/STAPFIA_0088_0129-0142.pdf
Also, this is not a correct citation.
Reference #31 is the exact copy of #44 with the same mistakes. Please adjust throughout the manuscript.
Reference #30, 29: I am not sure if these are cited correctly. Also, please provide concise translations in box brackets.
Reference #25: The year should not be in brackets.
Reference #17 and 14: I am not sure these are cited correctly.
Supplementary materials
This section is fine.
Authors contribution
Professor Hans ter Steege is missing from this section. Too bad, because I am positive he did add his knowledge to this well-executed and (mostly) well-written manuscript.
Acknowledgements – Funding – Data Availability Statement – Conflict of Interest
These sections are fine.
Author Response
We very much appreciate this thoughtful review that helped us correct several mistakes, and make the manuscript more cohesive and clear to readers.
We agreed with all suggested changes and made the modifications accordingly:
- we shortened the title
- replaced "know" by "information" in line 20
- added the suggested key words
- explained the ITS abbreviation in the text
- explained the terms primary and secondary forests in the introduction
- rewrote the sentence in line 59
- rewrote the last paragraph of the introduction about hypotheses (please see marked text in the manuscript)
- we replaced the "y-1". with "per year" or "yearly" throughout the manuscript
- in lines 123-125 we rewrote the sentence on plant families
- line 189: we could not find any word in bold near this line nor in any other parts of the main text
- the axes in Figure 4 are the CCA ordination axes without any other specific roles
- in the discussion, we linked the major findings to the hypotheses specified in the introduction
- we added the missing "the" in the conclusions
- we corrected all mistakes in the references and we are very grateful to the reviewer for spotting these
- in the authors' contribution section, we wrote full last name for easier identification and specified the involvement of Hans ter Steege.
Reviewer 3 Report
Very minor technical suggestions;
- Fig1; a b for the top four boxes, a1 b1 for the four boxes below
In the caption; Piro (P) Mogos (M)
- line 302 to 305 (Comparison ...) not clear sentence.
- Fig 3; ** what does it stand for
Author Response
Thank you for constructive comments. We made all suggested minor changes.
In Figure 1, instead of appending the Tukey HSD letters with numbers for each boxplot, we specified in the caption that these letters indicate significant differences within each boxplot.
We rewrote the sentence in lines 302-305
In the caption of Figure 3, we clarified that the and indicated significance levels of p<0.05 and p<0.01, respectively.
Round 2
Reviewer 2 Report
Dear Authors,
I am glad my review was of help to your team to improve the manuscript. I appreciate that you relied on my remarks during creating this version.
I have to reflect on two things.
Line 189:
Indeed, I can’t see anything in bold. It must have been a temporary error on my monitor, or within my Acrobat Reader software. It does happen at times.
Figure 4, ordination axes:
I accept the explanation. Bioinformatics is not my field, obviously.
It has been my pleasure to review this interesting piece of work. I wish you lots of citations J
Kind regards,
Your reviewer.